# *FBN1* Splice-Altering Mutations in Marfan Syndrome: A Case Report and Literature Review

**DOI:** 10.3390/genes13101842

**Published:** 2022-10-12

**Authors:** James Jiqi Wang, Bo Yu, Yang Sun, Xiuli Song, Dao Wen Wang, Zongzhe Li

**Affiliations:** 1Division of Cardiology, Departments of Internal Medicine and Genetic Diagnosis Center, Tongji Hospital, Tongji Medical College, Huazhong University of Science and Technology, Wuhan 430030, China; 2Hubei Key Laboratory of Genetics and Molecular Mechanisms of Cardiological Disorders, Huazhong University of Science and Technology, Wuhan 430030, China

**Keywords:** Marfan syndrome, *FBN1*, splicing mutation, whole-exome sequencing, genetic diagnosis

## Abstract

Marfan syndrome (MFS) is a life-threatening autosomal dominant genetic disorder of connective tissue caused by the pathogenic mutation of *FBN1*. Whole exome sequencing and Sanger sequencing were performed to identify the pathogenic mutation. The transcriptional consequence of the splice-altering mutation was analyzed via minigene assays and reverse-transcription PCR. We identified a novel pathogenic mutation (c.8051+1G>C) in the splice site of exon 64 of the *FBN1* gene in an MFS-pedigree. This mutation was confirmed to cause two different truncated transcripts (entire exon 64 skipping; partial exon 64 exclusion). We also systematically summarized previously reported transcriptional studies of pathogenic splice-altering mutations in the *FBN1* gene to investigate the clinical and transcriptional consequences. In conclusion, we reported for the first time that a splice-altering mutation in the *FBN1* gene leads to two abnormal transcripts simultaneously.

## 1. Introduction

Marfan syndrome (MFS) is a highly penetrant lethal autosomal dominant genetic disorder [1]. The estimated prevalence is about 6.5 per 100,000 individuals. The most significant clinical manifestations involve the skeletal and ocular systems, including ectopia lentis and skeletal abnormalities (tall stature, disproportionately long arms and legs, abnormally flexible joints, scoliosis) [2]. Regarding the cardiovascular system, patients often present with aortic root aneurysm and acute aortic dissection, accompanied by significantly decreased life expectancy (average age of death is 45 years) [2,3]. Patients with MFS are diagnosed mainly based on typical manifestations combined with genetic testing [4]. Many MFS patients are not diagnosed until they reach the stage of aortic dissection, which is a dangerous and lethal condition [1]. With early diagnosis and appropriate management, the life expectancy of individuals with MFS can approach that of the general population [2].

Pathogenic variants in *FBN1*, which encodes the important extracellular matrix glycoprotein fibrillin-1, are the main cause of MFS [5]. To date, over 1500 mutations have been identified in the *FBN1* gene worldwide (http://www.umd.be/FBN1, accessed on 10 August 2022) [5]. Splice-site altering mutation is one of the important mutation types identified that is associated with MFS, accounting for about 10% of mutations (HGMD database: http://www.hgmd.cf.ac.uk, accessed on 10 August 2022; ClinVar database: https://www.ncbi.nlm.nih.gov/clinvar/, accessed on 10 August 2022) [6].

Splice-site altering mutations can lead to a variety of transcriptional consequences, including exon skipping, exon partial deletion by cryptic splice site activation, intron retention, and pseudo exon inclusion. Therefore, the pathogenesis of MFS caused by each splice-site altering mutation needs to be individually analyzed [6].

Herein, we present a novel splice-altering pathogenic mutation (c.8051+1G>C) in the splice site of exon 64 of *FBN1* that causes MFS. Splicing analysis revealed that this mutation can cause both complete and partial (two splicing products) deletion of exon 64, which lead to pretermination and shortened fibrillin-1 protein.

## 2. Materials and Methods

### 2.1. Whole Exome Sequencing (WES)

The genomic DNA of the patient was obtained from peripheral blood obtained from the proband and his relatives using a QIAamp DNA Mini Kit (Qiagen, Hilden, Germany). The extracted DNA underwent electrophoresis to make sure there was no significant degradation or RNA pollution. To thoroughly uncover the genetic pathogenesis of the patient’s disease, WES was performed according to a previously established protocol [7].

### 2.2. Sanger Sequencing

Potentially pathogenic mutations and regions with low coverage were verified using Sanger sequencing on an Applied Biosystems 3500 sequencer (Applied Biosystems, Waltham, MA, USA). Subsequently, verified pathogenic mutations were sequenced in the patient’s relatives and 200 unrelated, healthy Chinese controls via Sanger sequencing.

### 2.3. Copy Number Variations (CNVs) Analysis

The CNVs of the proband were checked via a CNV workflow depending on the WES data [7]. We used quantitative real-time PCR analysis to validate potential pathogenic CNVs.

### 2.4. Cell Culture and Minigene Assays

Wild type and mutant minigene plasmids were constructed for the *FBN1* mutation (c.8051+1G>C) using the exon trap vector PET01 (MoBiTEx, Hannover, Germany). The sequence of Exon 64, part of intron 63, and part of intron 64 were amplified from the patient’s or his father’s genomic DNA and inserted into the minigene plasmid (PET01-FBN1). The following primers were used for target sequence amplification: forward 5′-taccgggccccccctcgagCCATGTTGGTTCATGACCGGAT-3′ and reverse 5′-cggtggcggccgctctagaTGACGTTTCCAGAAATCCAGATG-3′. Plasmids were then constructed and transfected into human embryonic kidney (HEK 293) cell lines in triplicates using Lipofectamine 2000 (Invitrogen, USA). Cells were cultured as we previously described [8].

### 2.5. RNA Extraction, cDNA Synthesis, and Splice Site Analysis

The total RNA was obtained 48 h after transfection with an RNeasy Mini Kit (Qiagen, Hilden, Germany). DNase I (Takara, Shiga, Japan) was then applied to the extracted RNA samples. RNA quality and concentration were tested via a NanoDrop 2000 spectrophotometer (Thermo Fisher Scientific, Waltham, MA, USA) and electrophoresis (UltraPure, Invitrogen, Waltham, MA, USA). RT-PCR was performed via a Primer Script^TM^ RT Reagent kit (Takara, Shiga, Japan).

For RT-PCR, we designed a pair of primers to amplify the target sequence derived from the expressed minigene: forward 5′- GATCGATCCGCTTCCTGCCCC-3′ and reverse 5′- TTCTGCCGGGCCACCTCCAG-3′. The primers were located on the vector sequences at each end of the PET01 Exon. Finally, the PCR product was confirmed by Sanger sequencing and visualized with electrophoresis on a 1% agarose gel.

### 2.6. Protein Structural Modeling

To predict the structural changes of the identified mutation, we obtained structural models of the wild-type and two mutant fibrillin-1 proteins using the AlphaFold Protein Structure Data Base (https://github.com/deepmind/alphafold, accessed on 14 August 2022) (DeepMind Technologies, Cambridge, UK). The predicted protein structures were visualized via a UCSF ChimeraX (RBVI, San Francisco, CA, USA).

### 2.7. Literature Retrieval

The splice-altering mutations of the *FBN1* gene in the ClinVar database (https://www.ncbi.nlm.nih.gov/clinvar/, accessed on 10 August 2022), HGMD database (http://www.hgmd.cf.ac.uk/ac/search.php, accessed on 10 August 2022), and UMD database (http://www.umd.be/FBN1, accessed on 10 August 2022) were systematically searched. First, we searched for ‘FBN1′ in the databases, with results limited to ‘splice site’. Afterwards, we went through all of the published studies of splice-altering mutations to those where functional analysis had been performed. Lastly, we summarized and recorded the reported experimental-based consequences and phenotypes of splice-altering mutations.

## 3. Results

### 3.1. Case Report

The proband was a 28-year-old male. His height was 185 cm with an increased arm span to height ratio and decreased upper segment to lower segment ratio. His head was not dolichocephalic, and no abnormal facial appearance was observed. He stated no history of pneumothorax. He had pectus carinatum and reduced elbow extension. His wrist sign was negative and his thumb sign was positive. He had lumbar scoliosis (Figure 1A), protrusio acetabuli, and pes planus. Regarding ocular symptoms, he did not have ectopia lentis but did have myopia (four diopters). An echocardiographic study showed no cardiac abnormalities. Computed tomography angiography (CTA) revealed multiple penetrating ulcers of the abdominalis, mild dilation of the aortic sinus, and arcus aortae. His father was diagnosed with dissection of the aorta at the age of 38. He also had similar systemic features and passed away due to aortic dissection rupture. His uncle and grandfather shared his systemic symptoms and passed away due to sudden cardiac death at the ages of 36 and 39, respectively (Figure 1B).

Genetic testing revealed a pathogenic *FBN1* splice-altering mutation in intron 64 (c.8051+1G>C) (Figure 2). Combining his dilated aortic sinus, positive systemic features (10 points), and family history, he was diagnosed with MFS according to the revised Ghent criteria for diagnosis of MFS [4].

### 3.2. Genetic Analysis

After WES and Sanger sequencing verification, we identified a potential pathogenic variation, c.8051+1G>C, in intron 64 of the gene *FBN1*. No other potential pathogenic variants or CNVs were discovered. The mutation was located in an evolutionarily highly conservative locus (Figure 2). The mutation was found to be co-segregated in the pedigree and was absent in 200 unrelated healthy controls. It was not detected in the gnomAD database. Additionally, it was absent from the databases of patients (ClinVar, https://www.ncbi.nlm.nih.gov/clinvar/, accessed on 10 August 2022; HGMD, http://www.hgmd.cf.ac.uk/ac/search.php, accessed on 10 August 2022). Therefore, according to the ACMG guideline [9], the identified mutation was classified as ‘pathogenic’ (PVS1+PM2+PP1+PP4).

### 3.3. Splicing Analysis

To further evaluate the impact of this splice-site mutation, minigene assays were used to investigate the transcriptional outcome of the identified mutation in *FBN1* (c.8051+1G>C). An FBN1-PET01 minigene was constructed (Figure 3A) and the cDNA of the wild-type and mutant mRNAs was obtained via RT-PCR (Figure 3B). The electrophoresis showed a strand around 500 bp in the lane of the wild-type cDNA, and two smaller strands in the lane of the mutant cDNA (Figure 3B).

Further Sanger sequencing of the mutant cDNA revealed two types of abnormal splicing products. The first product (Mutant type A) had a complete deletion of exon 64 (232 bp). The other product (Mutant type B) had a partial deletion of exon 64 (183 bp) (Figure 3C).

### 3.4. Protein Structural Modeling

Using protein structural modeling analysis, we compared the structural differences between the wild-type fibrillin-1 protein and the two types of mutant fibrillin-1 proteins. The mutant type A transcriptional product had a 232 bp frameshift deletion of the cDNA, causing the pretermination of protein (198 amino acids). The mutant type B transcriptional product had an in-frame 183 bp deletion of the cDNA, which leads to shortened fibrillin-1 protein (46 amino acids). Both mutant proteins showed significant structural changes compared to the wild-type protein (Figure 4).

## 4. Discussion

Via WES, Sanger sequencing, and minigene functional analysis, we linked a novel pathogenic splice site mutation (c.8051+1G>C) in *FBN1* to MFS, and further investigated the transcriptional pathogenesis that underlies this mutation. We report for the first time that a single splice-altering mutation in the *FBN1* gene can lead to two mutant transcripts simultaneously.

To acquire a further understanding of the transcriptional consequences of the identified mutation, a minigene analysis was performed. The minigene assay revealed that the novel splice site mutation (c.8051+1G>C) caused two types of deletion, including complete (Type A) and partial (Type B) exon 64 deletion. Type A is recognized as a 232 bp (frameshift) deletion which leads to a misreading of all the nucleotides downstream (67 mistranslated amino acids) and results in pretermination (198 amino acids shortened). Type B causes a 183 bp (in-frame) deletion resulting in shortened protein of 46 amino acids. Both structures of the mutant type fibrillin-1 protein show significant differences (Figure 4).

In all, three pathogenic mutations involving the splice site of exon 64 in the *FBN1* gene, with functional analysis, have been previously reported: c.8051+5G>A, c.8051+1G>A and c.8051+1G>T (also known as IVS64+1G>T). All three mutations were reported to link to either the classic MFS phenotype or an incomplete MFS phenotype, suggesting that mutations near this site can greatly disturb the functions of *FBN1*. For c.8051+5G>A, the carrier showed the classic MFS phenotype involving ocular, skeletal, and cardiac symptoms [10]. RNA sequencing analysis demonstrated that this mutation causes exon 64 skipping [10]. In addition to the three mutations, c.8051+5G>T has also been reported according to the HGMD database (www.hgmd.cf.ac.uk/ac/index.php, accessed on 10 August 2022), but the phenotype of this mutation is uncertain, and no functional analysis has been conducted.

Interestingly, in the same position of our mutation (c.8051+1G>C), two different pathogenic splice-altering mutations (c.8051+1G>A, c.8051+1G>T) have been reported to cause completely different clinical phenotypes. The first mutation, c.8051+1G>A, also reported to cause exon 64 skipping by RNA sequencing, was linked to the classic MFS phenotype [11]. However, the second mutation, c.8051+1G>T, was identified in a 3.5-year-old female patient (de novo) with a mixed phenotype of neonatal progeroid syndrome and MFS [12]. She was observed to have progeroid facial signs of neonatal onset, large head circumference with hydrocephaly, lipodystrophy, and tall stature [12]. Unfortunately, further functional analysis of this splice-altering mutation was unavailable in the report [12]. From the above analysis, it can be inferred that this site (c.8051+1) is an important functional splicing site of the *FBN1* gene. Different modes of mutation may lead to different transcriptional consequences and clinical phenotypes.

According to the ClinVar database, to date, there are 258 splice site mutations of *FBN1*, with the vast majority (226/258) of them being pathogenic or likely pathogenic, suggesting that splicing mutation is an important mechanism that leads to MFS. In addition, we summarized previously reported splice-altering mutations that had functional or transcriptional analysis from the ClinVar, HGMD, and UMD databases (Table 1). Out of 30 splicing *FBN1* mutations that had experimental analysis, 27 resulted in exome skipping (Table 1). Besides our mutation, only one splice-altering mutation caused partial exon deletion (c.4747+5G>T) [8]. Only one splice-altering mutation (c.3463+1G>A) resulted in intron retention and created a novel premature termination codon [13]. The detailed clinical features of the patients were described in 20 cases (Table 1). Out of 20 patients, 16 had symptoms involving all three systems (skeletal, ocular, and cardiovascular) and 19 out of 20 had cardiovascular symptoms. This suggests that a splice-altering mutation leads to a rather high incidence of cardiovascular abnormalities. As previously mentioned, our minigene assays indicated that c.8051+1G>C has two truncated transcriptional cDNA products (Type A and Type B). This result of multiple transcripts caused by a single splice-altering mutation was not observed in the *FBN1* gene (Table 1). However, it needs to be emphasized that cDNA products were obtained from the HEK-293 cell-line. The splicing result is from an in vitro pattern analysis and may not be completely in accordance with what takes place in the patient’s body. The specific mechanisms of how one splice-altering mutation causes multiple transcripts and the detailed clinical characteristics of patients carrying a mutation with multiple transcripts of *FBN1* need further investigation.

In conclusion, we identified a novel splice-altering pathogenic mutation in *FBN1* in a pedigree of MFS. Minigene assays revealed the previously unreported transcriptional consequences of this mutation. These findings enrich the pathogenic spectrum of *FBN1* and will help in genetic diagnosis and counseling in the future.

## Figures and Tables

**Figure 1 genes-13-01842-f001:**
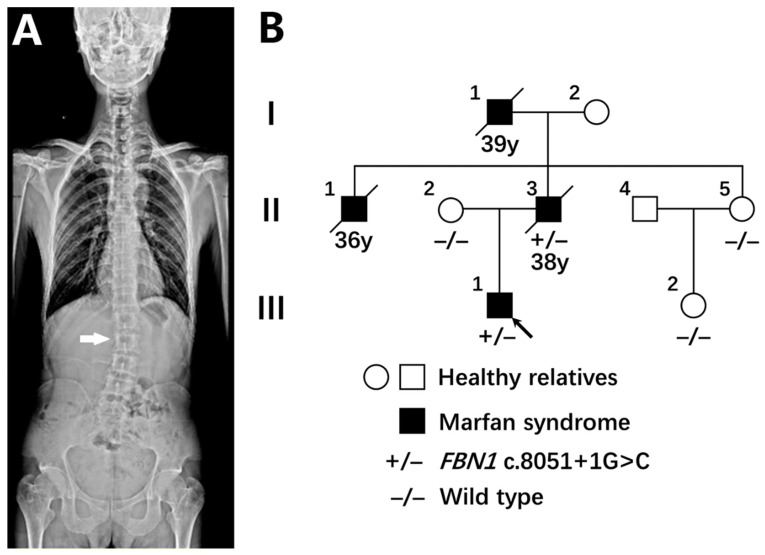
The clinical characteristics of the pedigree. (**A**) The anteroposterior X-ray of the spine of the proband revealed lumbar scoliosis. (**B**) The family tree and genotypes of the MFS pedigree. Arrow indicates the proband. Males and females are indicated by squares and circles, respectively. Black filled symbols represent MFS clinically affected individuals. +/−, represents heterozygous *FBN1* c.8051+1G>C mutation. −/−, represents wild type. Y, represents years old.

**Figure 2 genes-13-01842-f002:**
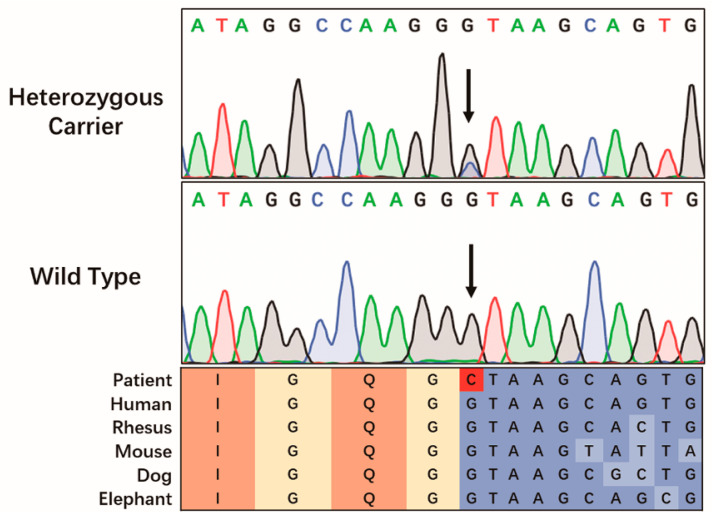
The Sanger sequencing results of the MFS pedigree. Arrows indicate the position of the mutation. Colored blocks show the evolutionary conservation around the mutation across multiple species.

**Figure 3 genes-13-01842-f003:**
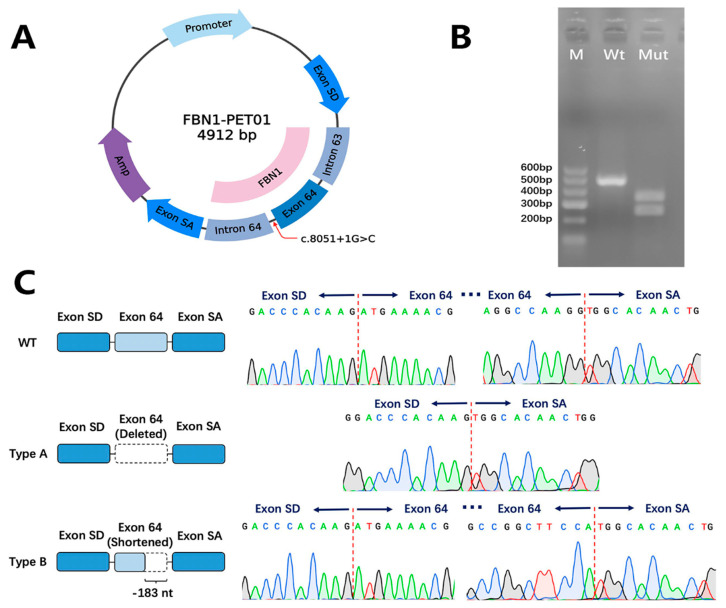
The results of the minigene assays of the splice-altering mutation. (**A**) The construction of the FBN1-PET01 minigene plasmid. (**B**) The electrophoresis result of the wild-type and mutant cDNA products. M represents marker. Wt represents cDNA from the wild-type plasmid. Mut represents cDNA from the mutant plasmid. (**C**) A schematic diagram and the Sanger sequencing results of the wild-type and two mutant cDNA products.

**Figure 4 genes-13-01842-f004:**
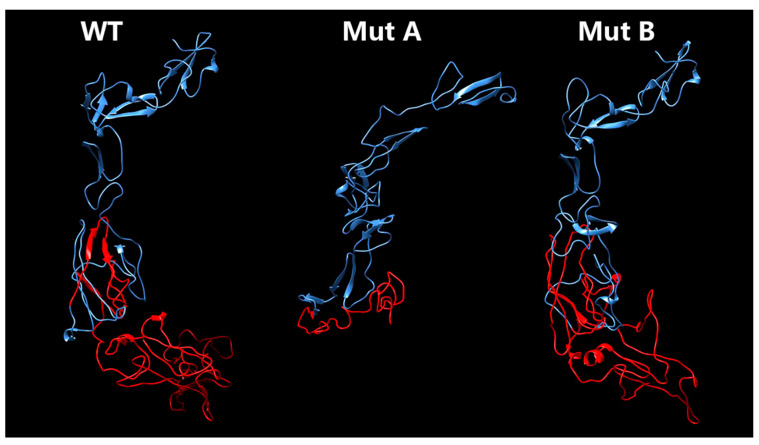
Protein structural modeling analysis of the wild-type and two mutant fibrillin-1 proteins. The amino acids before the mutation are indicated in blue. The amino acids after the mutation are indicated in red.

**Table 1 genes-13-01842-t001:** Summary of the experiment-based transcriptional consequences of splice-altering mutations of FBN1.

Mutation	Transcriptional Consequences	Phenotypes			References
		SS	OS	CS	
c.8052−2A>G	Skip of Exon 65	+	+	+	Hutchinson (2001) [14]
c.8051+5G>A	Skip of Exon 64	+	+	+	Hutchinson (2001) [14]
c.8051+1G>C	Skip of Exon 64/Deletion of 138bp	+	+	+	This Study
c.8051+1G>A	Skip of Exon 64	NA	NA	NA	Liu et al. (1997) [11]
c.7331−2A>G	Skip of Exon 60	+	+	+	Palz et al. (2000) [15]
c.7330+1G>A	Skip of Exon 59	+	+	+	Attanasio et al. (2008) [16]
c.7205−2A>G	Skip of Exon 59	+	+	+	Liu et al. (1996) [17]
c.6997+1G>A	Skip of Exon 57	+	+	+	Liu et al. (1996) [17]
c.6739+1G>C	Skip of Exon 55	+	+	+	Godfrey et al. (1993) [10]
c.6379−26C>T	Skip of Exon 52	NA	NA	NA	Liu et al. (1997) [18]
c.6164−1G>A	Skip of Exon 51	NA	NA	NA	Franken (2015) [19]
c.6163+2 del16bp	Skip of Exon 50	+	+	+	Liu et al. (1996) [17]
c.5917+6T>C	Skip of Exon 48	+	-	+	Wang et al. (2013) [20]
c.5788+5G>A	Skip of Exon 47	+	+	+	Liu et al. (1996) [17]
c.5788+1G>A	Insertion of 33 bp	+	+	+	Hutchinson (2001) [14]
c.4943−1G>C	Skip of Exon 41	NA	NA	NA	Liu et al. (1997) [11]
c.4817−2delA	Skip of Exon 40	+	+	+	Tjeldhorn (2015) [21]
c.4747+5G>T	Deletion of 48 bp in exon 38	+	-	-	McGrory et al. (1999) [22]
c.4459+1G>A	Skip pf Exon 36	NA	NA	NA	Liu et al. (1997) [11]
c.4087+1G>A	Skip pf Exon 33	+	-	+	Wang et al. (1995) [23]
c.3965−2A>T	Skip pf Exon 33	+	-	+	Wang et al. (1995) [23]
c.3964+1G>A	Skip of Exon 32	+	+	+	Booms et al. (1999) [24]
c.3963A>G	Skip of Exon 32	NA	NA	NA	Liu et al. (1997) [11]
c.3839−1G>T	Skip of Exon 32	NA	NA	NA	Liu et al. (1996) [17]
c.3463+1G>A	Insertion with PTC	+	+	+	Karttunen et al. (1998) [13]
c.3208+5G>T	Skip of Exon 26	+	+	+	Kainulainen et al. (1994) [25]
c.2854+1G>T	Skip of Exon 24	+	+	+	Tiecke et al. (2001) [26]
c.2293+2T>C	Skip of Exon 19	NA	NA	NA	Liu et al. (1996) [17]
c.1468+5G>A	Skip of Exon12	NA	NA	NA	Liu et al. (1997) [11]
c.247+1G>A	Skip pf Exon 3	NA	NA	NA	Dietz et al. (1993) [27]

SS, skeletal symptoms; OS, ocular symptoms; CS, cardiac symptoms; PTC, premature termination codon; NA, specific phenotype unavailable.

## Data Availability

The original contributions presented in this study are included in the article. Further inquiries can be directed to the corresponding author.

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
