# Peer review of "FBN1 Splice-Altering Mutations in Marfan Syndrome: A Case Report and Literature Review"

_genes, 2022, doi:10.3390/genes13101842_

Round 1

Reviewer 1 Report

The manuscript of Wang et al. represents a case study of the newly discovered FBN1 splice-altering mutation that causes Marfan syndrome and the review of all available literature on genetic causes of this disease. The study is well designed, structured, and described and can be of interest to medical doctors. My only recommendation is to check thoroughly the grammar of sentences.

For example (the correct writing is shown in red), in line 84: a pair of primers.

Line 105: no abnormal facial appearance was observed.

Line 109: no abnormal cardiac abnormalities.

Line 217: and 19 out of 20 had cardiovascular symptoms.

Line 222: how one splice-altering mutation causes multiple transcripts

Author Response

The manuscript of Wang et al. represents a case study of the newly discovered FBN1 splice-altering mutation that causes Marfan syndrome and the review of all available literature on genetic causes of this disease. The study is well designed, structured, and described and can be of interest to medical doctors. My only recommendation is to check thoroughly the grammar of sentences.

RESPONSE: Thank you very much for your valuable advice on this manuscript. We are sorry for the grammar mistakes. We have carefully checked our manuscript and corrected the grammar of sentences. 

Reviewer 2 Report

The authors describe the identification and characterization of a splicing mutation (c.8051+1G>C) in the FBN1 gene, identified in a patient with Marfan syndrome. From in vitro splicing pattern analysis, the mutation is associated with two aberrant transcripts (exon 64 skipping and partial intron deletion).

The study has some point of concern:

1-in the manuscript the mutation is reported as located in the 3' splice site of exon 64. However this is a mistake since, based on nomenclature, the mutation is located in the donor splice site of exon 64, which is the 5'ss.

2-page 5, lines 150-151: authors reported the splicing pattern analysis as Figure 3A, but is Figure 3B. Please correct.

3-in the discussion, authors reported that exon skipping is the most common outcome of FBN1 splicing mutations. This is obvious and should not be reported as new finding. 

4-authors did not discuss the tissue-specific splicing outcome. In this light, authors should discuss that what they are observing, it should be HEK-specific and not what happens in the patients.

5-authors did not comment why they exploited the HEK293 cell line to evaluate the splicing pattern.

Author Response

The authors describe the identification and characterization of a splicing mutation (c.8051+1G>C) in the FBN1 gene, identified in a patient with Marfan syndrome. From in vitro splicing pattern analysis, the mutation is associated with two aberrant transcripts (exon 64 skipping and partial intron deletion).

The study has some point of concern:

  1. in the manuscript the mutation is reported as located in the 3' splice site of exon 64. However this is a mistake since, based on nomenclature, the mutation is located in the donor splice site of exon 64, which is the 5'ss.

RESPONSE: Thank you for your comment. We noticed that, according to the nomenclature, c.8051+1G>C mains the mutation of the first base coming after the exon 64. This mutation located near the 3’ of exon 64, but in the 5’ of intron 64. Thank you again for mentioning this important issue. We are sorry that we did not make this clear enough. We have addressed this mistake in the revised manuscript.

  1. page 5, lines 150-151: authors reported the splicing pattern analysis as Figure 3A, but is Figure 3B. Please correct.

RESPONSE: Thanks for this important comment. We are sorry for this mistake. We have corrected it.

  1. in the discussion, authors reported that exon skipping is the most common outcome of FBN1 splicing mutations. This is obvious and should not be reported as new finding. 

RESPONSE: Thank you for your comment. We have deleted this sentence from the Discussion section (line 227-229).

  1. authors did not discuss the tissue-specific splicing outcome. In this light, authors should discuss that what they are observing, it should be HEK-specific and not what happens in the patients.

RESPONSE: This is a valuable advice. We have emphasized this in the Discussion section that this is an in vitro analysis and may not be completely in accordance with what happens in patients (line 236-238).

  1. authors did not comment why they exploited the HEK293 cell line to evaluate the splicing pattern.

RESPONSE: This is an important issue. HEK293 is a commonly used human-derived cell line for minigene analysis. We believe that human-derived cell lines can better mimic pathophysiological processes. Various published articles have used this cell line to perform minigene analysis[1-3].

Reviewer 3 Report

The authors reported a novel FBN1 splice-altering mutation in Marfan syndrome, which can expand the genetic spectrum and can help us better understand the genetic background of the disease. However:

1.       The title writes ‘……and systematic literature review’. Actually, it seems that the authors’ work was a narrative review rather than a systematic review. Please provide the detailed protocol in searching, screening, extracting and analyzing the data.

2.       There are more reported splice site mutations of FBN1 in HGMD than that the author analyzed in ‘Discussion’ section. Please clarify it.

3.       In addition to c.8051+5G>A, c.8051+1G>A, and c.8051+1G>T, c.8051+5G>T has also been reported in intron 64. I suggest the authors to make a thorough literature review, as I stated in comment 1.

4.       There are several typo, grammatical and punctuation mark errors throughout the manuscript, I suggest the manuscript be proofread by a native English speaker.

Author Response

The authors reported a novel FBN1 splice-altering mutation in Marfan syndrome, which can expand the genetic spectrum and can help us better understand the genetic background of the disease. However:

  1. The title writes ‘……and systematic literature review’. Actually, it seems that the authors’ work was a narrative review rather than a systematic review. Please provide the detailed protocol in searching, screening, extracting and analyzing the data.

RESPONSE: Thank you for this valuable advice. We have added detailed protocol regarding how the previously reported splice site mutations were summarized in the ‘Materials and Methods’ methods section (line 102-106).

  1. There are more reported splice site mutations of FBN1 in HGMD than that the author analyzed in ‘Discussion’ section. Please clarify it.

RESPONSE: Thank you very much for pointing this out. Indeed, there more reported splicing site mutations in HGMD database. However, Table 1 mainly focused on the experimental-based transcriptional consequences. We only summarized mutations for which functional experiments were well-performed. We have clarified this in the Discussion section (line 195-203).

  1. In addition to c.8051+5G>A, c.8051+1G>A, and c.8051+1G>T, c.8051+5G>T has also been reported in intron 64. I suggest the authors to make a thorough literature review, as I stated in comment 1.

RESPONSE: Thank you for your comment. We have also found this mutation in the HGMD database and have added it to the discussion section (line 200-203) right after we finished discussing section. Since the phenotype of the carrier of this mutation was uncertain and did not conduct functional analysis, we did not add this mutation to Table 1.

  1. There are several typo, grammatical and punctuation mark errors throughout the manuscript, I suggest the manuscript be proofread by a native English speaker.

RESPONSE: Thank you for pointing this out. We are sorry for the errors in the manuscript. We have asked for a native English speaker to proofread the manuscript, checked and revised the mistakes in the manuscript.

References:

  1. Xin, Q.; Liu, Q.; Liu, Z.; Shi, X.; Liu, X.; Zhang, R.; Hong, Y.; Zhao, X.; Shao, L. Twelve exonic variants in the SLC12A1 and CLCNKB genes alter RNA splicing in a minigene assay. Front Genet 2022, 13, 961384, doi:10.3389/fgene.2022.961384.
  2. Zhang, X.; Xie, Y.; Xu, K.; Chang, H.; Zhang, X.; Li, Y. Comprehensive Genetic Analysis Unraveled the Missing Heritability in a Chinese Cohort With Wolfram Syndrome 1: Clinical and Genetic Findings. Invest Ophthalmol Vis Sci 2022, 63, 9, doi:10.1167/iovs.63.10.9.
  3. Varela, M.D.; Bellingham, J.; Motta, F.; Jurkute, N.; Ellingford, J.M.; Quinodoz, M.; Oprych, K.; Niblock, M.; Janeschitz-Kriegl, L.; Kaminska, K.; et al. Multi-disciplinary team directed analysis of whole genome sequencing reveals pathogenic non-coding variants in molecularly undiagnosed inherited retinal dystrophies. Hum Mol Genet 2022, doi:10.1093/hmg/ddac227.

Round 2

Reviewer 2 Report

Authors do reply to all the raised points. I do not have any further concerns

Author Response

Thank you for your valuable advice on this manuscript. We really appreciate it.

Reviewer 3 Report

The manuscript is now more logical and readable. However,

1.       I still suggest to change ‘systematic literature review’ to ‘literature review’ in the title. A systematic review is defined as a review using a systematic method to summarize evidence on questions with a detailed and comprehensive plan of study. However, the review in this manuscript is descriptive and does not fulfill the criteria of a systematic review.

2.       Three still exist grammatical errors. Some examples: line 103-105, line 202, line 222, line 224.

Author Response

I still suggest to change ‘systematic literature review’ to ‘literature review’ in the title. A systematic review is defined as a review using a systematic method to summarize evidence on questions with a detailed and comprehensive plan of study. However, the review in this manuscript is descriptive and does not fulfill the criteria of a systematic review.

RESPONSE: Thank you for pointing this out. We have changed the title to ‘literature review’ instead of ‘systematic literature review’.

Three still exist grammatical errors. Some examples: line 103-105, line 202, line 222, line 224.

RESPONSE: We are sorry for the grammar mistakes. We have corrected the grammar mistakes.